# Character Level based Detection of DGA Domain Names

## Abstract

Recently several different deep learning architectures have been proposed that take a string of characters as the raw input signal and automatically derive features for text classification. Few studies are available that compare the effectiveness of these approaches for character based text classification with each other. In this paper we perform such an empirical comparison for the important cybersecurity problem of DGA detection: classifying domain names as either benign vs. produced by malware (i.e., by a Domain Generation Algorithm). Training and evaluating on a dataset with 2M domain names shows that there is surprisingly little difference between various convolutional neural network (CNN) and recurrent neural network (RNN) based architectures in terms of accuracy, prompting a preference for the simpler architectures, since they are faster to train and less prone to overfitting.

## 1 Introduction

Malware is software that infects computers in order to perform unauthorized malicious activities. In order to successfully achieve its goals, the malware needs to be able to connect to a command and control (C&C) center. To this end, both the controller behind the C&C center (hereafter called botmaster) and the malware on the infected machines can run a Domain Generation Algorithm (DGA) that generates hundreds or even thousands of domains automatically. The malware then attempts at resolving each one of these domains with its local DNS server. The botmaster will have registered one or a few of these automatically generated domains. For these domains that have been actually registered, the malware will obtain a valid IP address and will be able to communicate with the C&C center.

The binary text classification task that we address in this paper is: given a domain name string as input, classify it as either malicious, i.e. generated by a DGA, or as benign. Deep neural networks have recently appeared in the literature on DGA detection Woodbridge et al. (2016); Saxe & Berlin (2017); Yu et al. (2017). They significantly outperform traditional machine learning methods in accuracy, at the price of increasing the complexity of training the model and requiring larger datasets. Independent of the work on deep networks for DGA detection, other deep learning approaches for character based text classification have recently been proposed, including deep neural network architectures designed for processing and classification of tweets (Dhingra et al. (2016); Vosoughi et al. (2016)) as well as general natural language text (Zhang et al. (2015)). No systematic study is available that compares the predictive accuracy of all these different character based deep learning architectures, leaving one to wonder which one works best for DGA detection.

To answer this open question, in this paper we compare the performance of five different deep learning architectures for character based text classification (see Table 1) for the problem of detecting DGAs. They all rely on character-level embeddings, and they all use a deep learning architecture based on convolutional neural network (CNN) layers, recurrent neural network (RNN) layers, or a combination of both. Our most important finding is that for DGA detection, which can be thought of as classification of short character strings, despite of vast differences in the deep network architectures, there is remarkably little difference among the methods in terms of accuracy and false positive rates, while they all comfortably outperform a random forest trained on human engineered features. This finding is of practical value for the design of deep neural network based classifiers for short text classification in industry and academia: it provides evidence that one can select an architecture that

Table 1: High level overview of recent deep learning approaches for character based text classification

| Name | Architecture | Reference |
|------|-------------|-----------|
| Endgame | single LSTM layer | Woodbridge et al. (2016) |
| Invincea | parallel CNN layers | Saxe & Berlin (2017) |
| CMU | forward LSTM layer + backward LSTM layer | Dhingra et al. (2016) |
| MIT | stacked CNN layers + single LSTM layer | Vosoughi et al. (2016) |
| NYU | stacked CNN layers | Zhang et al. (2015) |

is faster to train, without loss of accuracy. In the context of DGA detection, optimizing the training time is of particular importance, as the models need to be retrained on a regular basis to stay current with respect to new, emerging malware.

## 2    BACKGROUND

Malware controllers, or botmasters, use malware for all kinds of unauthorized malicious activities. These activities range from stealing information, to exploiting the victims' computing resources to mine bitcoin. They can also include launching a distributed denial of service attack from the victims's computers or encrypting the victims hard drive (ransomware). In order to successfully achieve its goals, it is vital that the malware be able to connect to a command and control (C&C) center. This communication can serve many purposes. The malware can use it to send stolen information (such as passwords or access credentials) to the malware designer behind the C&C center, it can use this communication channel to receive instructions or even to update itself to a newer version.

Initially, botmasters established such a communication channel to the C&C center by hard-coding an IP address inside the malware. This approach has obvious shortcomings from the botmasters' perspective: once the malware is reversed engineered, the IP address is discovered and shut down. Over time, malware designers came up with a much more effective strategy: Domain Generation Algorithms (DGAs). Domain Generation Algorithms work by having the malware accessing some available source of randomness and inputting it into an algorithm that generates hundreds or even thousands of domains automatically. The malware then attempts at resolving each one of these domains with its local DNS server (a DNS server runs a protocol that translates domain names into IP addresses; it is a vital piece of the Internet). The botmaster will have registered one or a few of these automatically generated domains. For these domains that have been actually registered, the malware will obtain a valid IP address and will be able to communicate with the C&C center. For all the other domains that were automatically generated but not registered, the malware obtains a message stating that these domains could not have been resolved and ignores them.

DGAs make blacklisting of domains extremely difficult, since by changing the initial randomness (while keeping the same algorithm) the malware can potentially generate completely different domains. This technique has been used by high-profile malware such as Conficker, Stuxnet (the malware designed to attack Iran nuclear facilities) and Flame. Catching domain names generated by malware has become a central topic in information security, leading to a recent interest in detecting DGA domains using machine learning techniques. Models that classify domain names as benign or malicious based solely on the domain name string are of particular interest for their generality, as context information beyond the domain name string might be unavailable or expensive to acquire. Traditional machine learning methods for DGA detection based on the domain name string rely on extraction of predefined, human engineered lexical features, see e.g. Antonakakis et al. (2012); Schiavoni et al. (2014). Whenever human engineered features are used, it is obvious that this opens the door for an adversary to carefully craft its DGA to avoid detection by using the aforementioned features. This makes maintaining such machine learning systems labor intensive. Recently proposed deep learning techniques for detecting DGAs learn features automatically, thereby offering the potential to bypass the human effort of feature engineering (Woodbridge et al. (2016); Saxe & Berlin (2017); Yu et al. (2017)). In addition these deep learning approaches outperform the traditional machine learning techniques with human engineered features in terms of accuracy and false positive rates. The choice of deep network architecture in these recent works appears fairly arbitrary, and it

is unclear whether they would perform better or worse than other deep neural networks approaches for text classification that have been introduced recently, in particular character-level methods for processing and classification of tweets (Dhingra et al. (2016); Vosoughi et al. (2016)) and for general natural language text (Zhang et al. (2015)). In this paper, we provide the first comparative study of all these different methods, showing that, despite of vast differences in the architectures, they can be easily tuned to result in the same predictive accuracy.

The problem of adversarial examples, i.e. instances that have intentionally been designed to cause the model to make a mistake, are a well known problem in machine learning. The scenario sketched above — in which a malware designer exploits knowledge about the lexical features used by a random forest to craft his DGA to avoid detection — is a prime example of this. Deep neural networks are famously not immune to adversarial examples either, and generative adversarial networks (GANs) can be trained to generate them automatically (see e.g. Goodfellow et al. (2014a;b)). Specifically in the context of DGA detection, Anderson et al. (2016) have used a character-based generative adversarial network (GAN) to augment training sets in order to harden other machine learning models (like a random forest) against yet-to-be-observed DGAs. It is highly unlikely for attackers to use GANs themselves, because DGA algorithms must be light enough to be embedded inside malware code. Furthermore, generating domain names that look like a benign domain is not enough for an effective DGA. Ideally, every domain produced by a DGA must not have been registered yet or must have a low likelihood of being registered already – if a domain produced by a DGA has already been taken, it is useless for the botmaster. Combining all these requirements is essential for a serious study of adversarial generated domains and outside the scope of this paper.

## 3 METHODS

We compare five different deep learning methods for short string classification, when applied to the problem of DGA detection specifically. For each of the methods, we start from the original proposals as can be found in the references in Table 1 and only make modifications when they improve the predictive accuracy for the classification of domain names. Below we give an overview of the methods and the adaptations made. A Keras[1] code snippet for each method is included in Appendix A.

The strings that we give as input to all classifiers consist of a second level domain (SLD) and a top level domain (TLD), separated by a dot, as in e.g. *wikipedia.org*. Following Woodbridge et al. (2016), we set the maximum length at 75 characters, padded with zeros on the left for domains whose length is less than 75.[2] We convert each domain name string to lower case, since domain names are case insensitive, and encode it as an ASCII code sequence of length 128, effectively representing each domain name string as a 75 by 128 matrix in which each character corresponds to a column.

### 3.1 RNN BASED ARCHITECTURES

**Endgame Model**    Long short-term memory networks (LSTMs), a special kind of recurrent neural networks (RNNs) have recently attracted a lot of attention because of their successful application to problems that involve processing of sequences (Hochreiter & Schmidhuber (1997)). Since domain names can be thought of as sequences of characters, LSTMs are a natural kind of classifiers to apply. The LSTM network proposed by Woodbridge et al. (2016) was designed specifically for DGA detection, so we stay very close to the original model. The network is comprised of an embedding layer, an LSTM layer (128 LSTM cells with default Tanh activation), and a single node output layer with sigmoid activation. Instead of using RMSProp as the optimization algorithm, as was done in Woodbridge et al. (2016), we switched to Adam (Kingma & Ba (2014)) because it resulted in better loss convergence results (see Section 4). The Endgame model includes dropout, a technique to improve model performance and overcome over-fitting by randomly excluding nodes during training,

---

[1] `https://github.com/fchollet/keras`, Accessed: 2017-05-28
[2] The maximum allowed length for SLDs and TLDs is 63 characters each. In practice they are typically shorter. The longest domain name string we encountered in our experiments is 73 characters. This string includes the SLD, the TLD, and the dot that separates them.

which serves to break up complex co-adaptations in the network Srivastava et al. (2014). This is confined to the training phase; all nodes are active during testing and deployment.

The role of the embedding layer is to learn to represent each character that can occur in a domain name by a 128-dimensional numerical vector. This vector is different from the original 128-dimensional ASCII encoding. The embedding maps semantically similar characters to similar vectors, where the notion of similarity is implicitly derived (learned) based on the classification task at hand. As will become clear in the remainder of this section, all five deep neural network architectures under study start with such an embedding layer. To allow for a fair comparison, we have made the parameter choices for the embedding layer, such as the dimensionality of the embedding space, identical for all five models. In addition, for comparison purposes, in Section 4 we also present the results of a "baseline neural network model" consisting *only* of an embedding layer as its hidden layer.

**CMU Model**  Bidirectional RNNs extend regular RNNs by processing the input string in two ways. In a forward layer, the input sequence is processed from the left to the right, as in a traditional RNN, while in a backward layer, the processing happens from the right to the left. The output from the forward and the backward layer is then combined and passed on to further layers. Bidirectional LSTMs for character level text processing have been proposed in Ling et al. (2015), and, following up on that, very similar bidirectional GRUs (gated recurrent units) have been applied in a "Tweet2Vec" model for tweet classification (predicting hashtags of tweets) by Dhingra et al. (2016). We use an adaptation of the latter; see Listing 2 in Appendix A. Including dropout or replacing LSTM by GRU did not cause a significant change in predictive accuracy, although the latter did result in a decrease of training runtime.

## 3.2  CNN BASED ARCHITECTURES

**NYU Model**  Convolutional neural networks (CNNs) are known for their ability to process input data with a grid like topology, such as images consisting of a grid of pixels. To the best of our knowledge, Zhang et al. (2015) were the first to apply 1-dimensional or "temporal" CNNs successfully to text classification at character level. Their proposed deep network architecture, which is intended to process full-blown natural language text such as news articles or reviews, includes 6 stacked CNN layers, with each subsequent layer consuming the output from the previous layer. In contrast to natural language text, domain names are very short and they do not have an internal grammatical structure, naturally resulting the original architectures from Zhang et al. (2015) to overfit on our data. We therefore reduced the number of stacked CNN layers to two, and decreased the size and the number of filters on the CNN layers (see Listing 3).

**Invincea Model**  Saxe & Berlin (2017) proposed a CNN based classifier that takes generic short character strings as its input and learns to detect whether they are indicators of malicious behavior. The short character strings can be e.g. URLs, file paths, or registry keys. The fundamental difference between the Invincea model versus the NYU model described above, is that in the Invincea model the CNN layers are parallel instead of stacked, and that pooling always happens over the entire domain name instead of within a small pooling window. That means that the Invincea model is only detecting the presence or absence of patterns in the domain names, and does not retain any information on where exactly in the domain name string these patterns occur. In the Invincea model, the embedding layer is followed by a convolutional layer with 1024 filters, namely 256 filters for each of the sizes 2, 3, 4, and 5. Each of these filters learns to detect the soft presence of an interesting soft $n$-gram (with $n = 2, 3, 4, 5$). The output of the convolutional layer is consumed by two dense hidden layers, each with 1024 nodes, before reaching a single node output layer with sigmoid activation. Out of all the models that we compared, this one has the most extensive architecture.

## 3.3  HYBRID CNN/RNN BASED ARCHITECTURE

**MIT Model**  The MIT model proposed by Vosoughi et al. (2016) is an extension of the NYU model, where the stacked CNN layers are followed by an LSTM layer. Similarly as with the NYU model, the use of multiple stacked CNN layers (which worked well for tweets in Vosoughi et al. (2016)) resulted in the models to overfit on our data. For this reason, we reduced the MIT model architecture to the minimum that preserves its spirit: one CNN layer followed by one LSTM layer.

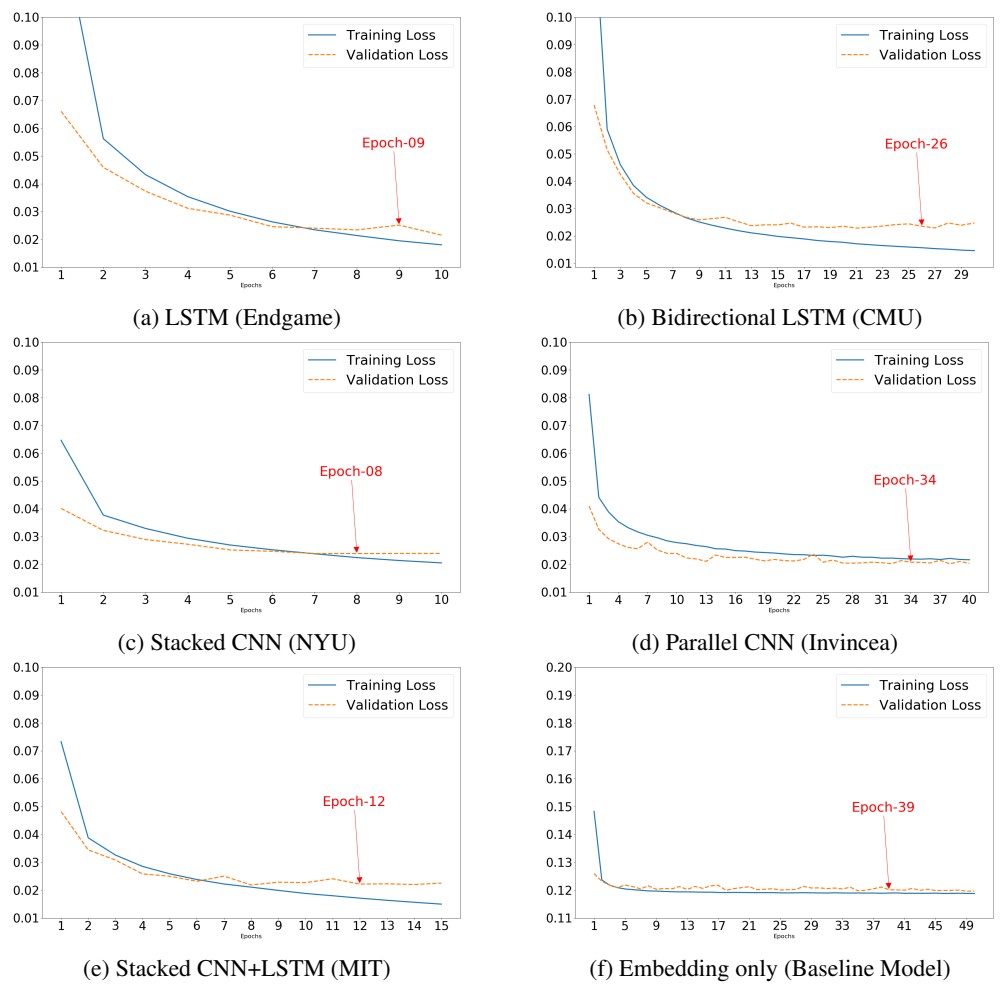

Figure 1: Training and validation loss curves. The vertical axis in Figure (f) has a different scale than the other figures, related to the fact that the "Embedding only" baseline model has a much higher loss than the other models.

## 4 RESULTS AND LEARNED REPRESENTATIONS

We trained and evaluated the models on a dataset with 1 million DGA domain names from Bambenek[3] (positive examples) and the top 1 million domain names from Alexa[4] (negative examples). Alexa ranks websites based on their popularity in terms of number of page views and number of unique visitors. It only retains the websites' SLD and TLD, aggregating across any subdomains. For example, according to Alexa, the five highest ranked domain names in terms of popularity on 2017-10-26 are *google.com*, *youtube.com*, *facebook.com baidu.com*, and *wikipedia.org*. For our experiments, we assume that the top 1 million domain names in this ranking are benign domain names, although it is possible that the bottom of the ranking may contain some noise.

In addition to these benign domain names, we collected 1 million DGA domain names from the Bambenek Consulting feeds for 3 different days, namely Jun 24, Jul 22, Jul 23, 2017. These feeds contain DGA domain names from specific malware families that were observed in real traffic on those days. Such domain names can be collected by reverse engineering a known malware family, generating lists of domain names with the reverse engineered malware, and checking which of these domain names also occur in real traffic. Note that our goal in this paper is the development of

---

[3] http://osint.bambenekconsulting.com/feeds/, Accessed 2017-07-23
[4] https://www.alexa.com, Accessed 2017-05-28

Table 2: Results on test data from July 2017. Accuracy, TPR, FPR are w.r.t. a threshold that gives a FPR of 0.001 on the validation data.

| Model | Architecture | | Acc | TPR | FPR | AUC@1% |
|---|---|---|---|---|---|---|
| RF | Lexical features | | 91.51% | 83.15% | 0.00128 | 84.77% |
| MLP | Lexical features | | 73.74% | 47.61% | 0.00091 | 58.81% |
| Embedding | | | 84.29% | 68.69% | 0.00108 | 80.88% |
| | LSTM | CNN | | | | |
| Endgame | x | | 98.72% | 97.55% | 0.00102 | 98.03% |
| Invincea | | x | 98.95% | 98.01% | 0.00109 | 97.47% |
| CMU | x | | 98.54% | 97.18% | 0.00108 | 98.25% |
| MIT | x | x | 98.70% | 97.49% | 0.00099 | 97.55% |
| NYU | | x | 98.58% | 97.27% | 0.00116 | 97.93% |

a neural network classifier that can detect DGAs without the need to reverse engineer malware families. An important advantage of such a classifier is that it can also be used against new and previously unknown malware families.

We randomly split the data into 80% for training, 10% for validation, and 10% for testing. Figure 1 shows the training and validation loss curves for each of the models described in Section 3. The displayed epochs indicate where we stopped the training to obtain the models used to produce the final results in Table 2 and 5. The training loss is higher than the validation loss in the pictures in Figure 1 because the loss against the training data is computed in an average way across batches (the batch size is 100) while dropout is being applied, whereas performance on the validation set is determined at the end of each epoch with dropout disabled. Figure 1(f) displays the loss curves for training a simple neural network consisting of only an embedding layer. We include the performance of this network in our results as a baseline.

The accuracy of each of the trained models when applied to the test data is recorded in Table 2. In addition to accuracy, this table includes the true positive rate (TPR) and false positive rate (FPR) for each of the models. Recall that TPR = TP/(TP+FN) and FPR = FP/(FP+TN) where TP, FP, TN, and FN are the number of true positives, false positives, true negatives, and false negatives respectively. A low false positive rate is very important in deployed DGA detection systems, because blocking legitimate traffic is highly undesirable. All classifiers in Table 2 output a probability that a given instance belongs to the positive class, so we can tune a threshold probability at which to consider a prediction positive. For each model, we choose this threshold such that the model trained over the training data has a 0.001 FPR over the validation data. Then we report the accuracy, TPR and FPR obtained with this classification threshold over the test data. Finally, we also report AUC@1%FPR, which is the integral of the ROC curve from FPR = 0 to FPR = 0.01 on the test data.

For comparison purposes, Table 2 also contains results for a Random Forest (RF) and a Multilayer Perceptron (MLP) trained on the following 11 features, extracted from each domain name string (see Yu et al. (2014; 2016)): ent (normalized entropy of characters); nl2 (median of 2-gram); nl3 (median of 3-gram); naz (symbol character ratio); hex (hex character ratio); vwl (vowel character ratio); len (domain label length); gni (gini index of characters); cer (classification error of characters); tld (top level domain hash); dgt (first character digit). The Random Forest consists of 100 trees. The MLP has a single hidden layer with 128 nodes; see Listing 7 in Appendix A. Like the deep networks in this paper, this MLP was trained with batch size 100. The values of the 11 features are normalized so that they are all on the same scale before presenting them to the MLP.

As expected, the FPR of all classifiers is around 0.001. There is a clear variation in the TPR that the classifiers achieve against that small FPR. While the Random Forest is only able to "catch" 83% of the malicious domain names, all the deep neural network architectures achieve a recall of 97-98%. The baseline neural network consisting of only an embedding layer as its hidden layer clearly performs the worst with a TPR of less than 69%, highlighting that it is advantageous to extend the network architecture with one or more LSTM or CNN layers. Interestingly, there is little to no variation among the five deep neural network architectures in terms of TPR.

Table 3 contains examples of domain names that were randomly selected among those misclassified by either the Random Forest or by all five deep neural networks. Inspecting the column of the

Table 3: Examples of domain names that were either misclassified by the Random Forest or by the deep neural networks. For the malicious domain names, the name of the malware family is shown between parentheses.

|  | benign | malicious |
|---|---|---|
| misclassified by RF & correctly classified by all five deep networks | kosmetikosdnr.lt
jobrankingcommittee.com
naturalandhealthytips.com
pokemonrubysapphire.com
turkcehdpornoizle.com
bollywoodparksdubai.com | mowvcssclilpomqi.com (murofet)
ntearasildeafeninguvuc.com (banjori)
raklloblmuppono.info (cryptolocker)
5alo1ch3wvn5o1cc.org (chinad)
ldjucxqhivnaperisusb.ga (necurs)
daeontibyxgask.cc (ranbyus) |
| misclassified by all five deep networks & correctly classified by RF | rfembassy.kz
4553t5pugtt1qslvsnmpc0tpfz5fo.xyz
9odyefoccu1gririlemjijbab.top
a5rtngpo9840oyd.com
mydwnldsghtfv.com | doycsnramt.com (qakbot)
gypjuytopleh.com (ramnit)
zamdazhocs.com (nymaim)
mxdsbbnxmogo.online (tinba)
pelkbazgro.info (pykspa) |

benign domain names, i.e. the Alexa domain names, it is interesting to note that most of those misclassified by the deep neural networks (bottom left in the table) come across as gibberish that a human annotator would likely also classify as malicious. As also evident from the top right of Table 3, the deep neural networks have become very good at considering such gibberish-looking domain names to be malicious, even though they were never explicitly told to do so (unlike the Random Forest, which explicitly includes a normalized entropy of characters feature). The fact that the malicious domain names at the bottom right of Table 3 were missed by the deep neural networks might be due to our deliberate choice to tune the classification threshold to achieve a very low FPR. This makes all the classifiers hold back from labeling a domain name as malicious if they are not almost completely certain. As explained above, a low FPR is very important in deployed DGA detection systems, as blocking legitimate traffic is highly undesirable. Note that if a deployed DGA detection system would rely on the Random Forest classifier, it would block all domain names from the first row in Table 3, whereas, if it would rely on any of the deep neural network classifiers, it would block all domain names from the second row in Table 3. The domain names from the first column would have been unjustly blocked. For those negatively affected by this, it would be easier to "understand" (and perhaps forgive) the decisions made by the deep neural network classifiers, as they are more in line with decisions that a human would make when confronted with these domain name strings.

The results in Table 2 are for DGA domain names that appeared at the same time in real traffic as the training data used to construct the classifiers, namely July 2017. To evaluate how well the trained models hold up against DGA domain names that appear at a later point, we collected an additional 100K DGA domain names from the Bambenek Consulting feeds for Dec 6, Dec 9, and Dec 10, 2017. Table 4 contains results for this "prospective" dataset. The only difference between the experimental setup for Table 2 and Table 4 are the 100K DGA domain names used in the test set. The training dataset, the validation dataset, and the 100K Alexa domain names used in the test set are kept the same. As can be seen when comparing the results in Table 2 and Table 4, the to be expected drop in TPR is minor.

As Table 5 shows there is a substantial distinction among the different models in terms of complexity (number of parameters that have to be learned during the training process) and required training time per epoch. The platform used for training is an AWS virtual machine with access to multiple GPUs. Both the number of epochs needed to train a network, and the number of seconds required per epoch, are contributing factors to the overall training runtime. The NYU model took less than 8 minutes to train, while the CMU model took as much as 10 hours. The last column in Table 5 shows the time needed to classify 200K domain names with an already trained model. The ranking of the deep networks in terms of this "scoring time" coincides with the ranking in terms of training time. Given that all deep networks achieve a similar accuracy (TPR), the NYU model with it short training and scoring time comes out as the winner.

Table 4: Results on test data from December 2017. Accuracy, TPR, FPR are w.r.t. a threshold that gives a FPR of 0.001 on the validation data.

| Model | Architecture | | Acc | TPR | FPR | AUC@1% |
|---|---|---|---|---|---|---|
| RF | Lexical features | | 91.57% | 83.26% | 0.00128 | 84.47% |
| MLP | Lexical features | | 78.87% | 57.86% | 0.00091 | 67.23% |
| Embedding | | | 86.75% | 73.61% | 0.00108 | 82.93% |
| | **LSTM** | **CNN** | | | | |
| Endgame | x | | 98.22% | 96.54% | 0.00102 | 97.53% |
| Invincea | | x | 98.44% | 97.00% | 0.00109 | 97.07% |
| CMU | x | | 98.06% | 96.23% | 0.00108 | 97.46% |
| MIT | x | x | 98.21% | 96.52% | 0.00099 | 97.52% |
| NYU | | x | 98.12% | 96.35% | 0.00116 | 97.10% |

Table 5: Comparison of complexity and efficiency of classifiers for DGA detection. The complexity refers to the number of parameters that have to be learned in the deep learning architectures. The training time is reported in terms of seconds needed for an epoch times the number of epochs. The scoring time is the time in seconds needed to label 200K domain names by an already trained model.

| Model | Architecture | | Complexity | Training Time | | Scoring time |
|---|---|---|---|---|---|---|
| RF | Lexical features | | 100 trees | | 1,800s | 5s |
| MLP | Lexical features | | 1,665 par | 10s × 40 = | 400s | 1s |
| Embedding | | | 25,985 par | 15s × 40 = | 600s | 3s |
| | **LSTM** | **CNN** | | | | |
| Endgame | x | | 148,097 par | 430s × 10 = | 4,300s | 13s |
| Invincea | | x | 2,576,385 par | 105s × 40 = | 4,200s | 7s |
| CMU | x | | 115,329 par | 1200s × 30 = | 36,000s | 26s |
| MIT | x | x | 115,137 par | 800s × 15 = | 12,000s | 10s |
| NYU | | x | 254,337 par | 45s × 10 = | 450s | 5s |

## 5 CONCLUSION

DGA detection, i.e. the classification task of distinguishing between benign domain names and those generated by malware (Domain Generation Algorithms), has become a central topic in information security. In this paper we have compared five different deep neural network architectures that perform this classification task based purely on the domain name string, given as a raw input signal at character level. All five models, i.e. two RNN based architectures, two CNN based architectures, and one hybrid RNN/CNN architecture perform equally well, catching around 97-98% of malicious domain names against a false positive rate of 0.001. This roughly means that for every 970 malicious domain names that the deep networks catch, they flag only one benign domain name erroneously as malicious. A Random Forest based on human defined linguistic features achieves a recall of only 83% against the same 0.001 false positive rate when trained and tested on the same data that was used for the deep networks. The use of a deep neural network that automatically learns features is attractive in a cybersecurity setting because it is a lot harder to craft malware to avoid detection by a system that relies on automatically learned features instead of on human engineered features. An interesting direction for future work is to test the trained deep networks more extensively on domain names generated by new and previously unseen malware families.

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

## A  KERAS CODE FOR DEEP NETWORKS

```
main_input = Input(shape=(75, ), dtype='int32', name='main_input')
embedding = Embedding(input_dim=128, output_dim=128, input_length=75)(main_input)
lstm = LSTM(128, return_sequences=False)(embedding)
drop = Dropout(0.5)(lstm)
output = Dense(1, activation='sigmoid')(drop)
model = Model(inputs=main_input, outputs=output)
model.compile(loss='binary_crossentropy', optimizer='adam')
```

Listing 1: Endgame model with single LSTM layer, adapted from Woodbridge et al. (2016)

```
main_input = Input(shape=(75, ), dtype='int32', name='main_input')
embedding = Embedding(input_dim=128, output_dim=128, input_length=75)(main_input)
bi_lstm = Bidirectional (layer=LSTM(64, return_sequences=False),
                         merge_mode='concat')(embedding)
output = Dense(1, activation='sigmoid')(bi_lstm)
model = Model(inputs=main_input, outputs=output)
model.compile(loss='binary_crossentropy', optimizer='adam')
```

Listing 2: CMU model with bidirectional LSTM, adapted from Dhingra et al. (2016)

```
main_input = Input(shape=(75, ), dtype='int32', name='main_input')
embedding = Embedding(input_dim=128, output_dim=128, input_length=75)(main_input)
conv1 = Conv1D(filters=128, kernel_size=3, padding='same', strides=1)(embedding)
thresh1 = ThresholdedReLU(1e−6)(conv1)
max_pool1 = MaxPooling1D(pool_size=2, padding='same')(thresh1)
conv2 = Conv1D(filters=128, kernel_size=2, padding='same', strides=1)(max_pool1)
thresh2 = ThresholdedReLU(1e−6)(conv2)
max_pool2 = MaxPooling1D(pool_size=2, padding='same')(thresh2)
flatten = Flatten ()(max_pool2)
fc = Dense(64)(flatten)
thresh_fc = ThresholdedReLU(1e−6)(fc)
drop = Dropout(0.5)(thresh_fc)
output = Dense(1, activation='sigmoid')(drop)
model = Model(inputs=main_input, outputs=output)
model.compile(loss='binary_crossentropy', optimizer='adam')
```

Listing 3: NYU model with stacked CNN layers, adapted from Zhang et al. (2015)

```
def getconvmodel(self, kernel_size, filters):
    model = Sequential ()
    model.add(
        Conv1D(filters = filters, input_shape=(128, 128), kernel_size = kernel_size,
               padding='same', activation='relu', strides=1))
    model.add(Lambda(lambda x: K.sum(x, axis=1), output_shape=(filters, )))
    model.add(Dropout(0.5))
    return model

main_input = Input(shape=(75, ), dtype='int32', name='main_input')
embedding = Embedding(input_dim=128, output_dim=128, input_length=75)(main_input)
conv1 = getconvmodel(2, 256)(embedding)
conv2 = getconvmodel(3, 256)(embedding)
conv3 = getconvmodel(4, 256)(embedding)
conv4 = getconvmodel(5, 256)(embedding)
merged = Concatenate ()([conv1, conv2, conv3, conv4])
middle = Dense(1024, activation='relu')(merged)
middle = Dropout(0.5)(middle)
middle = Dense(1024, activation='relu')(middle)
```

```
middle = Dropout(0.5)(middle)
output = Dense(1, activation ='sigmoid')(middle)
model = Model(inputs=main_input, outputs=output)
model.compile(loss=' binary_crossentropy ', optimizer='adam')
```

Listing 4: Invincea CNN model with parallel CNN layers, adapted from Saxe & Berlin (2017)

```
main_input = Input(shape=(75, ), dtype='int32 ', name='main_input')
embedding = Embedding(input_dim=128, output_dim=128, input_length =75)(main_input)
conv = Conv1D( filters =128, kernel_size =3, padding='same', activation ='relu ',
     strides =1)(embedding)
max_pool = MaxPooling1D(pool_size=2, padding='same')(conv)
encode = LSTM(64, return_sequences=False)(max_pool)
output = Dense(1, activation ='sigmoid')(encode)
model = Model(inputs=main_input, outputs=output)
model.compile(loss=' binary_crossentropy ', optimizer='adam')
```

Listing 5: MIT model with a stacked CNN and LSTM layer, adapted from Vosoughi et al. (2016)

```
main_input = Input(shape=(75, ), dtype='int32 ', name='main_input')
embedding = Embedding(input_dim=128, output_dim=128, input_length =75)(main_input)
flatten = Flatten ()(embedding)
output = Dense(1, activation ='sigmoid')( flatten )
model = Model(inputs=main_input, outputs=output)
print (model.summary())
model.compile(loss=' binary_crossentropy ', optimizer='adam')
```

Listing 6: Baseline Model with only Embedding Layer

```
main_input = Input(shape=(11, ), name='main_input')
dense = Dense(128, activation ='relu ')(main_input)
output = Dense(1, activation ='sigmoid')(dense)
model = Model(inputs=main_input, outputs=output)
print (model.summary())
model.compile(loss=' binary_crossentropy ', optimizer='adam')
```

Listing 7: MLP Model with 128 Nodes Dense Layer

