# OpenReview forum: "Character Level Based Detection of DGA Domain Names"
_ICLR.cc/2018/Conference — Reject_

### Official Review · AnonReviewer2 · 2017-11-27
**A well-written paper that compares five deep architectures for the cybersecurity task of domain generation algorithm detection**

**Rating:** 7
**Confidence:** 4

**Review:**


SUMMARY

This paper addresses the cybersecurity problem of domain generation algorithm (DGA)  detection. A class of malware uses algorithms to automatically generate artificial domain names for various purposes, e.g. to generate large numbers of rendezvous points. DGA detection concerns the (automatic) distinction of actual and artificially generated domain names. In this paper, a basic problem formulation and general solution approach is investigated, namely that of treating the detection as a text classification task and to let domain names arrive to the classifier as strings of characters. A set of five deep learning architectures (both CNNs and RNNs) are compared empirical on the text classification task. A domain name data set with two million instances is used for the experiments. The main conclusion is that the different architectures are almost equally accurate and that this prompts a preference of simpler architectures over more complex architectures, since training time and the likelihood for overfitting can potentially be reduced.

COMMENTS

The introduction is well-written, clear, and concise. It describes the studied real-world problem and clarifies the relevance and challenge involved in solving the problem. The introduction provides a clear overview of deep learning architectures that have already been proposed for solving the problem as well as some architectures that could potentially be used. One suggestion for the introduction is that the authors take some of the description of the domain problem and put it into a separate background section to reduce the text the reader has to consume before arriving at the research problem and proposed solution.

The methods section (Section 2) provides a clear description of each of the five architectures along with brief code listings and details about whether any changes or parameter choices were made for the experiment. In the beginning of the section, it is not clarified why, if a 75 character string is encoded as a 128 byte ASCII sequence, the content has to be stored in a 75 x 128 matrix instead of a vector of size 128. This is clarified later but should perhaps be discussed earlier to allow readers from outside the subarea to grasp the approach.

Section 3 describes the experiment settings, the results, and discusses the learned representations and the possible implications of using either the deep architectures or the “baseline” Random Forest classifier. Perhaps, the authors could elaborate a little bit more on why Random Forests were trained on a completely different set of features than the deep architectures? The data is stated to be randomly divided into training (80%), validation (10%), and testing (10%). How many times is this procedure repeated? (That is, how many experimental runs were averaged or was the experiment run once?).

In summary, this is an interesting and well-written paper on a timely topic. The main conclusion is intuitive. Perhaps the conclusion is even regarded as obvious by some but, in my opinion, the result is important since it was obtained from new, rather extensive experiments on a large data set and through the comparison of several existing (earlier proposed) architectures. Since the main conclusion is that simple models should be prioritised over complex ones (due to that their accuracy is very similar), it would have been interesting to get some brief comments on a simplicity comparison of the candidates at the conclusion.

MINOR COMMENTS

Abstract: “Little studies” -> “Few studies”

Table 1: “approach” -> “approaches”

Figure 1: Use the same y-axis scale for all subplots (if possible) to simplify comparison. Also, try to move Figure 1 so that it appears closer to its inline reference in the text.

Section 3: “based their on popularity” -> “based on their popularity”

---

> ### Author Response · Authors · 2017-12-14
> **Additional test results on prospective data and comparison of computational performance added**
>
> We would like to thank reviewer 3 for the careful review provided to us, and respond to the comments below.
>
> The random forest classifier used in the experiments was trained on a set of expert defined features that is commonly used in machine learning models for DGA detection. In the revised version of the paper, we have also added results for a multilayer perceptron with 1 hidden layer that is trained on the same expert defined features as the random forest. These features are extracted from the domain names in a preprocessing step, i.e. each domain name is converted into a list of features (numerical values). The drawback of this is that it makes machine learning models for DGA detection vulnerable to being outdated, when hackers learn about the expert defined features and come up with new DGA algorithms to circumvent them. A recent, successful approach in machine learning for DGA detection is therefore to not extract any features at all a-priori and instead give the entire domain name string as a raw input signal to a deep network that learns by itself which features to extract. The features extracted by a deep network are in no way predefined by a human, and it is even famously difficult to interpret them. Furthermore, such deep networks can even outperform systems that incorporate human knowledge, as is clear from our experiments: all deep networks significantly outperform the random forest as well as a multilayer perceptron trained with human defined features. Note that there is no straightforward way at all to train a random forest directly on a raw domain name string. The ability to learn what features to extract from raw input is a defining characteristic of deep learning.
>
> For the results reported in the paper, we split the dataset once into 80% for training, 10% for validation, and 10% for testing, and kept this exact same split throughout all the experiments. In other words, the numbers reported in the paper result from running each experiment once. They are not averages. We are aware that for small to medium-sized datasets, k-fold cross-validation gives a more reliable estimate of the predictive performance of models. In our initial experiments (not included in the paper), we trained and evaluated some of the deep networks using 5-fold cross-validation. We found that the difference across folds was small, which is a known phenomenon when working with large datasets. Given the high computational cost for training an individual deep network (up to 10 hours), we therefore decided to go forward with a single 80-10-10 split and keep this consistent across all experiments, i.e. each model reported on in the paper is trained on exactly the same set of domain names, and tested on exactly the same set of domain names (fully disjoint from the training dataset). For the revised version of the paper, we have created an additional prospective test dataset with DGAs that were observed in real traffic in December 2017. The additional results included in the paper (Table 4) show how the deep neural networks trained on data from July 2017 hold up against such a prospective dataset with “future” data.
>
> Our results were obtained, as pointed out by reviewer 3, by computationally expensive experiments, carefully designed and implemented. They required computational power that can be prohibitive for many other researchers who are interested in using deep learning models for DGA detection.  As we previously stated, we believe our results are of practical value for people designing such kind of classifiers in industry and academia. In the revised version of the paper, we have extended our comparison of all the studied models regarding computational performance for training and scoring (as suggested by reviewer 3).
>
> Finally, we thank reviewer 3 for raising the presentation issues. We have addressed them in the revised version (including the creation of a separate “Background” section).

---

### Official Review · AnonReviewer3 · 2017-11-27
**Technically sound but little innovation/insight w.r.t. models**

**Rating:** 5
**Confidence:** 3

**Review:**

This paper applies several NN architectures to classify url’s between benign and malware related URLs.
The baseline is random forests and feature engineering.

This is clearly an application paper.
No new method is being proposed, only existing methods are applied directly to the task.

I am not familiar with the task at hand so I cannot properly judge the quality/accuracy of the results obtained but it seems ok.
For evaluation data was split randomly in 80% train, 10% test and 10% validation. Given the amount of data 2*10**6 samples, this seems sufficient.
I think the evaluation could be improved by using malware URLs that were obtained during a larger time window.
Specifically, it would be nice if train, test and validation URLs would be operated chronologically. I.e. all train url precede the validation and test urls.
Ideally, the train and test urls would also be different in time. This would enable a better test of the generalization capabilities in what is essentially a continuously changing environment.

This paper is a very difficult for me to assign a final rating.
There is no obvious technical mistake  and the paper is written reasonably well.
There is however a lack of technical novelty or insight in the models themselves.
I think that the paper should be submitted to a journal or conference in the application domain where it would be a better fit.

For this reason, I will give the score marginally below the acceptance threshold now.
But if the other reviewers argue that the paper should be accepted I will change my score.

---

> ### Author Response · Authors · 2017-12-14
> **Additional test results on prospective data have been added**
>
> We would like to thank reviewer 2 for the careful review provided to us, and respond to the comments below.
>
> Deep Neural Networks have recently appeared in the literature on DGA detection. They significantly outperform traditional machine learning methods in accuracy, at the price of increasing the complexity of training the model and requiring larger datasets. While it is exciting to see yet another task where deep learning comes to the scene as a leading technique, the proposed methods were entirely arbitrary. There was no justification for the proposed architectures or the size of the networks and no clue on how much better these models would perform given some extra fine-tuning.
>
> We aimed at filling this gap in the literature. We collected several models previously used for text classification problems and DGA detection, optimized these models and compared them systematically and rigorously. We ended up showing that all these networks (after several optimizations and fine-tuning) performed equally well despite their vast differences. So, one should pick up the model that can be trained in the least amount of time and requires less data (has fewer parameters to be trained).
>
> We believe our results are robust and will be of practical value for people designing such kind of classifiers in industry and academia. Given that, in practice, these models would be continuously re-trained to add new families (online learning), optimizing the training time is an important question.
>
> The remark by reviewer 2 that a chronological split between training, validation, and test data would be “a better test of the generalization capabilities in a continuously changing environment” is a valid one. For the revised version of the paper, we have created an additional prospective test dataset with DGAs that were observed in real traffic in December 2017. The additional results included in the paper (Table 4) show how the deep neural networks trained on data from July 2017 hold up against such a prospective dataset with “future” data.

---

> > ### Comment · AnonReviewer3 · 2018-01-12
> > **-**
> >
> > I appreciate the effort to include the additional experiments.
> >
> > The positive points of this paper remain the correct technical evaluation and the multiple models being evaluated.
> > Technically the work appears to be solid and improved in the revision thanks to the additional experiments.
> >
> > Unfortunately, given the limited novelty, the paper remains borderline to me.

---

### Official Review · AnonReviewer1 · 2017-11-27
**-**

**Rating:** 4
**Confidence:** 4

**Review:**

This paper proposes to automatically recognize domain names as malicious or benign by deep networks (convnets and RNNs) trained to directly classify the character sequence as such.


Pros

The paper addresses an important application of deep networks, comparing the performance of a variety of different types of model architectures.

The tested networks seem to perform reasonably well on the task.


Cons

There is little novelty in the proposed method/models -- the paper is primarily focused on comparing existing models on a new task.

The descriptions of the different architectures compared are overly verbose -- they are all simple standard convnet / RNN architectures.  The code specifying the models is also excessive for the main text -- it should be moved to an appendix or even left for a code release.

The comparisons between various architectures are not very enlightening as they aren’t done in a controlled way -- there are a large number of differences between any pair of models so it’s hard to tell where the performance differences come from. It’s also difficult to compare the learning curves among the different models (Fig 1) as they are in separate plots with differently scaled axes.

The proposed problem is an explicitly adversarial setting and adversarial examples are a well-known issue with deep networks and other models, but this issue is not addressed or analyzed in the paper. (In fact, the intro claims this is an advantage of not using hand-engineered features for malicious domain detection, seemingly ignoring the literature on adversarial examples for deep nets.) For example, in this case an attacker could start with a legitimate domain name and use black box adversarial attacks (or white box attacks, given access to the model weights) to derive a similar domain name that the models proposed here would classify as benign.


While this paper addresses an important problem, in its current form the novelty and analysis are limited and the paper has some presentation issues.

---

> ### Author Response · Authors · 2017-12-14
> **Comparative study fills important gap in the literature on deep nets for DGA detection**
>
> We would like to thank reviewer 1 for the careful review provided to us, and respond to the comments below.
>
> Deep Neural Networks have recently appeared in the literature on DGA detection. They significantly outperform traditional machine learning methods in accuracy, at the price of increasing the complexity of training the model and requiring larger datasets. While it is exciting to see yet another task where deep learning comes to the scene as a leading technique, the proposed methods were entirely arbitrary. There was no justification for the proposed architectures or the size of the networks and no clue on how much better these models would perform given some extra fine-tuning.
>
> We aimed at filling this gap in the literature. We collected several models previously used for text classification problems and DGA detection, optimized these models and compared them systematically and rigorously. We ended up showing that all these networks (after several optimizations and fine-tuning) performed equally well despite their vast differences. So, one should pick up the model that can be trained in the least amount of time and requires less data (has fewer parameters to be trained).
>
> We believe our results are robust and will be of practical value for people designing such kind of classifiers in industry and academia. Given that, in practice, these models would be continuously re-trained to add new families (online learning), optimizing the training time is an important question.
>
> The issue raised by reviewer 1 about generating adversarial examples is an important one. In the revised version of the paper, we cited the following recent work, and put it in context:
>
> Anderson, Hyrum S., Jonathan Woodbridge, and Bobby Filar. "DeepDGA: Adversarially-Tuned Domain Generation and Detection." In Proceedings of the 2016 ACM Workshop on Artificial Intelligence and Security, pp. 13-21. ACM, 2016.
>
> In this paper, a character-based generative adversarial network (GAN) is used to augment training sets in order to harden other machine learning models (like a random forest) against yet-to-be-observed DGAs. It is highly unlikely for attackers to use GANs themselves, because DGA algorithms must be light enough to be embedded inside malware code. Furthermore, generating domain names that look like a benign domain is not enough for an effective DGA. Ideally, every domain produced by a DGA must not have been registered yet or must have a low likelihood of being registered already – if a domain produced by a DGA has already been taken, it is useless for the botmaster.  Combining all these requirements is essential for a serious study of adversarial generated domains and requires a paper of itself.
>
> The controlling factor in our experiments is, in a sense, the accuracy (TPR). The Endgame and Invincea models resulted, after very small adaptations, in a very similar accuracy. We adjusted the other architectures (CMU, NYU, MIT) to achieve a similar performance. This was relatively easy to do, with a good amount of trial and error. For all models, at 97%-98% TPR we hit a plateau through which we have not been able to break yet.
>
> We addressed the presentation issues raised by reviewer 1 in the revised version. The scale of the vertical axes in Figure 1 has been made consistent, and the Keras code snippets have been moved to an appendix.

---

### Decision · Program_Chairs · 2018-01-29
**ICLR 2018 Conference Acceptance Decision**

**Decision:**

Reject

**Comment:**

meta score: 4

This is basically an application in which some different deep learning approaches are compared on the task of automatically identifying domain names automatically generated by malware.  The experiments are well-constructed and reported.  However, the work does not have novelty beyond the application domain, and thus is not really suitable for ICLR.

Pros
 - good set of experiments carried out on an important task
 - clearly written
Cons
 - lacks technical novelty